# Adaptations to nitrogen availability drive ecological divergence of chemosynthetic symbionts

Isidora Morel-Letelier[1], Benedict Yuen[1], A. Carlotta Kück[1], Yolanda E. Camacho-García[2,3,4], Jillian M. Petersen[5], Minor Lara[6], Matthieu Leray[7], Jonathan A. Eisen[8,9], Jay T. Osvatic[10,11], Olivier Gros[12], Laetitia G. E. Wilkins[1]*

1 Eco-Evolutionary Interactions Group, Max Planck Institute for Marine Microbiology (MPIMM), Bremen, Germany, 2 Centro de Investigación en Ciencias del Mar y Limnología (CIMAR), Universidad de Costa Rica, San Pedro, San José, Costa Rica, 3 Centro de Investigación en Biodiversidad y Ecología Tropical (CIBET), Universidad de Costa Rica, San Pedro, San José, Costa Rica, 4 Escuela de Biología, Universidad de Costa Rica, San Pedro, San José, Costa Rica, 5 Centre for Microbiology and Environmental Systems Science, University of Vienna, Vienna, Austria, 6 Diving Center Cuajiniquil, Provincia de Guanacaste, Cuajiniquil, Costa Rica, 7 Smithsonian Tropical Research Institute, Balboa, Ancon, Republic of Panamá, 8 Department of Evolution and Ecology, University of California, Davis, Davis, California, United States of America, 9 Department of Medical Microbiology and Immunology, University of California, Davis, Davis, California, United States of America, 10 Joint Microbiome Facility of the Medical University of Vienna and the University of Vienna, Vienna, Austria, 11 Department of Laboratory Medicine, Medical University of Vienna, Vienna, Austria, 12 Institut de Systématique, Evolution, Biodiversité (ISYEB), Muséum National d'Histoire Naturelle, CNRS, Sorbonne Université, Université des Antilles, Pointe-à-Pitre, France

☯ These authors contributed equally to this work.

* lwilkins@mpi-bremen.de

**Data Availability Statement:** The raw read sets of the MAGs that were retrieved in this study were deposited in the NCBI BioProject PRJNA1010110 with Biosample accession Numbers SAMN38052512 - SAMN38052649 and

## Abstract

Bacterial symbionts, with their shorter generation times and capacity for horizontal gene transfer (HGT), play a critical role in allowing marine organisms to cope with environmental change. The closure of the Isthmus of Panama created distinct environmental conditions in the Tropical Eastern Pacific (TEP) and Caribbean, offering a "natural experiment" for studying how closely related animals evolve and adapt under environmental change. However, the role of bacterial symbionts in this process is often overlooked. We sequenced the genomes of endosymbiotic bacteria in two sets of sister species of chemosymbiotic bivalves from the genera *Codakia* and *Ctena* (family Lucinidae) collected on either side of the Isthmus, to investigate how differing environmental conditions have influenced the selection of symbionts and their metabolic capabilities. The lucinid sister species hosted different *Candidatus* Thiodiazotropha symbionts and only those from the Caribbean had the genetic potential for nitrogen fixation, while those from the TEP did not. Interestingly, this nitrogen-fixing ability did not correspond to symbiont phylogeny, suggesting convergent evolution of nitrogen fixation potential under nutrient-poor conditions. Reconstructing the evolutionary history of the *nifHDKT* operon by including other lucinid symbiont genomes from around the world further revealed that the last common ancestor (LCA) of *Ca.* Thiodiazotropha lacked *nif* genes, and populations in oligotrophic habitats later re-acquired the *nif* operon through HGT from the *Sedimenticola* symbiont lineage. Our study suggests that HGT of the *nif* operon has facilitated niche diversification of the globally distributed *Ca.* Thiodiazotropha

SAMN40439061 - SAMN40439241. The MAGs themselves, all the scripts, and a detailed description of all bioinformatic analyses performed can be found in the GitLab repo https://gitlab.mpi-bremen.de/imorel/isthmus-sym-evolution. Alignments and phylogenetic trees were made available on Figshare https://figshare.com/projects/ISTHMUS_SYM_EVOLUTION/18061.

**Funding:** This project received funding from the Max-Planck-Gesellschaft to LGWE. ACK, BY, LGWE and IML received salaries from Max-Planck-Gesellschaft. Part of the sequencing was carried out by the DNA Technologies and Expression Analysis Core at the UC Davis Genome Center, supported by NIH Shared Instrumentation Grant 1S10OD010786-01. JMP's and JTO's contributions were supported by the ERC Starting Grant EvoLucin (grant number 802494), and a Vienna Research Grant for Young Investigators from the Vienna Science and Technology Fund (WWTF, VRG14-021). LGEW was supported with salary and a postdoctoral fellowship by a Marie Curie individual postdoctoral fellowship "MSCA-IF-EF-RI" for project #Pansymbiosis with grant number SEP-210693430. We thank the Gordon and Betty Moore Foundation (https://www.moore.org) through Grant GBMF560 to JAE. The funders had no role in study design, data collection and analysis, decision to publish, or preparation of the manuscript.

**Competing interests:** The authors have declared that no competing interests exist.

endolucinida species clade. It highlights the importance of nitrogen availability in driving the ecological diversification of chemosynthetic symbiont species and the role that bacterial symbionts may play in the adaptation of marine organisms to changing environmental conditions.

## Author summary

Approximately three million years ago, the closure of the Isthmus of Panama connected North and South America, leading to species interchange on land but splitting an ancient ocean into the Tropical Eastern Pacific (TEP) and the Caribbean Sea. Today, these two marine habitats are characterized by significantly different environmental conditions. Notably, the Caribbean Sea became highly oligotrophic which caused a massive extinction event.

Our focus on bivalve species pairs that survived on both sides aimed at understanding how their associated bacterial symbionts enabled them to adapt to this massive environmental change. Although both Caribbean and TEP bivalves host *Candidatus* Thiodiazotropha symbionts, only those on the Caribbean side are capable of nitrogen fixation. This capability does not align with symbiont evolutionary history, indicating convergent evolution due to similar environmental pressures.

Exploring the genetic history of lucinid symbionts across the globe revealed that the ancestor of *Ca.* Thiodiazotropha lacked nitrogen fixation genes. Populations in nutrient-poor habitats acquired it multiple times through horizontal gene transfer (HGT). Our research underscores the role of HGT in bacterial adaptation and highlights the impact of nitrogen availability on symbiont ecological diversification. It shows how bacterial symbionts can aid marine organisms in adapting to environmental change.

## Introduction

Global change gives rise to new environmental conditions and niches that organisms can adapt to and exploit, and the extent and mechanisms of adaptation in marine species may be significantly influenced by their microbiomes [1]. Bacterial symbionts of animals can potentially adapt to changing environments more rapidly and more flexibly than their hosts due to traits such as shorter generation times, enhanced recombination capabilities, and the potential for horizontal gene transfer (HGT) between distantly-related organisms [2]. In addition, horizontally acquired bacterial symbionts—those obtained from the environment in each generation—have access to a larger genetic pool for genetic exchange during their free-living phase, which can lead to faster adaptive responses [3]. Consequently, symbionts can serve as a source of ecological innovation, enabling the symbiosis to tap into novel resources and adapt to novel habitats [4–6]. Understanding the mechanisms enabling microbial symbionts to adapt to new environmental conditions will provide novel insights into how animal-microbe symbioses respond to changing environments.

The closure of the Isthmus of Panama about 2.8 million years ago had a profound impact on oceanic conditions, altering environmental factors such as ocean currents, salinity, temperature, and nutrient availability on both sides of the Isthmus [7]. The Tropical Eastern Pacific (TEP) continued to experience regular nutrient input due to seasonal upwelling, coupled with increased primary productivity, variable temperatures, and strong tides [7]. In contrast, the

Caribbean coast became characterized by stable and warmer temperatures, higher salinity, and a notably low availability of organic nutrients [7]. Animal populations that were once connected became separated by the closure of the Isthmus, ultimately resulting in the emergence of sister species on separate evolutionary trajectories, diverging in response to the markedly different environmental conditions on either side of the Isthmus [8]. The adaptation strategies enabling these sister species to thrive in their respective, contrasting environments have been extensively studied, but these studies have primarily focused on the animals themselves, largely overlooking the potential influence of host-associated microorganisms (reviewed in [1,8,9]). The Isthmus of Panama presents a unique opportunity to investigate the drivers of diversification and adaptation through a "natural experiment" running for millions of years with a taxonomically replicated set of animal-microbe assemblages [10]. This offers valuable insights into the interplay between hosts and their microbial symbionts in the context of environmental change.

The Lucinidae, one of the most species-rich bivalve families, thrive in a wide array of marine environments and is the most diverse group of chemosymbiotic animals [11]. Lucinids house endosymbiotic sulfide-oxidizing Gammaproteobacteria intracellularly within specialized gill cells, where they use the energy derived from oxidizing reduced sulfur compounds to synthesize organic carbon [11–13]. This partnership is obligate for lucinids because they rely on their symbionts for a significant portion of their carbon nutritional requirements [14,15]. The bacterial symbionts are acquired from free-living populations in the environment during the early developmental stages of each new generation [16,17]. Recent studies have begun to unveil the metabolic and genomic diversity among symbionts from vastly different environments across the globe, including differences in their abilities to metabolize carbon and inorganic nitrogen [18–20]. Notably, some symbiont clades of the genus *Ca.* Thiodiazotropha were the first documented example of nitrogen-fixing chemosynthetic symbionts [21,22]. However, the precise processes governing the diversification and adaptation of lucinid symbionts to changing environmental parameters, which may lead to the divergence of local populations of globally distributed taxa, remains poorly understood. Sister species of lucinids from the genera *Codakia* and *Ctena* have diverged on either side of the Isthmus of Panama [1,11]. This unique relationship allows for a comparative study of symbiont adaptation, free from the confounding effects of host evolutionary history.

We used high-throughput metagenomic sequencing to recover metagenome-assembled genomes (MAGs) of bacterial symbionts associated with *Codakia* and *Ctena* sister species from both sides of the Isthmus of Panama. Our primary objective was to investigate how differing environmental conditions on either side of the Isthmus have influenced the selection, diversity, and functional traits of symbionts associated with lucinid clams. We found that all lucinid symbionts from the Caribbean had the potential to fix nitrogen and assimilate nitrate, but these functions were absent in all symbionts from the TEP. To further explore the evolutionary origins of nitrogen fixation in different lucinid symbiont lineages, we compared the genomes of symbionts across the Isthmus of Panama with other lucinid symbiont genomes from around the world. Using phylogenetic reconciliation, we reconstructed the evolutionary history of the *nifHDKT* operon and identified two distinct HGT events that led to *nif* gene acquisition and correlated with the colonization of nutrient-poor environments. Within the globally distributed *Ca.* Thiodiazotropha endolucinida clade, populations in nutrient-poor environments possessed nitrogen fixation genes, whereas those closely related populations in nutrient-rich environments did not. Despite their high Average Nucleotide Identity (ANI; >95%) and evidence of homologous recombination between geographically distant populations, our findings indicate that these *Ca.* Thiodiazotropha endolucinida populations have diverged ecologically and occupy separate niches that differ in nitrogen availability. We

hypothesize that the diversification of these symbionts was facilitated by the acquisition of the nitrogen fixation genes through HGT. Our results provide valuable insights into the dynamic interplay between environmental factors and genetic exchange that shape the ecology, evolution, and diversification of host-associated microorganisms.

## Results

### Comparing bacterial symbionts in host sister species across the Isthmus of Panama reveals complex phylogenetic relationships and differences in genomic potential

**Symbiont clades are exclusive to either side of the Isthmus and are independent of host taxonomy.** We recovered a total of 148 high-quality gammaproteobacterial MAGs from the gill metagenomes of *Codakia* and *Ctena* sister species sampled from nine different locations across the Isthmus: three sites in the TEP and six in the Caribbean (CAR; Fig 1A and 1B). Thirteen of these MAGs were retrieved from nine *Codakia orbicularis* (CAR) metagenomes, four from four *Codakia distinguenda* (TEP) metagenomes, 44 from 38 *Ctena imbricatula* (CAR) metagenomes, 44 from 31 *Ctena* sp. "COSTE" (CAR) metagenomes, and 43 from 37 *Ctena* cf. *galapagana* (TEP) metagenomes (metadata and statistics available in S1 and S2 Tables).

The MAGs constituted six distinct clades—three from each side of the Isthmus, but all were taxonomically assigned to the genus *Ca*. Thiodiazotropha (Fig 1C and 1D). Two clades were identified as previously described Caribbean lucinid symbionts *Ca*. T. taylori and *Ca*. T. endolucinida [19,22]. Three of the remaining four clades are previously undescribed symbiont species, based on an ANI threshold of 95% for species delimitation [24–26] (S1 Dataset). We discovered one new bacterial species clade from the Caribbean (*Ca*. Thiodiazotropha fergusoni), for which we propose the name after Walter Ferguson, a Panamanian-born calypso singer and songwriter based in Cahuita, Costa Rica (1919–2023). We designate the two new species clades from the TEP as *Ca*. Thiodiazotropha larai and *Ca*. Thiodiazotropha boucheti. These names honor Minor Lara for his contributions to marine research and conservation in the Guanacaste region of Costa Rica, and Dr. Philippe Bouchet for his extensive work on lucinids. The third clade from the TEP had an ANI of ~95.5% to *Ca*. T. endolucinida, suggesting its inclusion within this species, but it formed a distinct monophyletic sub-clade unique to the TEP. We shall hereafter refer to this clade as *Ca*. T. endolucinida TEP to distinguish it from the originally described Caribbean clade, which we will hereafter refer to as *Ca*. T. endolucinida CAR. Two of the MAGs classified as *Ca*. T. fergusoni and one classified as *Ca*. T. endolucinida TEP, from the samples sequenced with PacBio, were circularized. No symbiont clade was present on both sides of the Isthmus (Fig 1). However, unlike their hosts, the symbionts on either side of the Isthmus did not share a sister lineage relationship, indicating the absence of co-diversification in the host and symbiont phylogenies (Fig 1B and 1C). MAGs classified as *Ca*. T. fergusoni, *Ca*. T. endolucinida CAR and *Ca*. T. endolucinida TEP were recovered from both *Codakia* and *Ctena* specimens, while *Ca*. T. taylori, *Ca*. T. boucheti and *Ca*. T. larai MAGs were only recovered from *Ctena* specimens (Fig 1C and 1D).

**Symbiont clades across the Isthmus differ mainly in nitrogen metabolic capabilities.** We compared the metabolic potential of the symbionts from the TEP and Caribbean to identify genomic adaptations to the different environmental conditions across the Isthmus of Panama (Fig 1B). Core metabolic capabilities were shared among all clades (as shown in S2 Dataset) and with previously described members of *Ca*. Thiodiazotropha [19,27]. All clades possessed genes responsible for sulfur oxidation and carbon fixation via the Calvin Cycle. Notably, *Ca*. T. fergusoni MAGs only encoded the RubisCo type I, while *Ca*. T. larai and

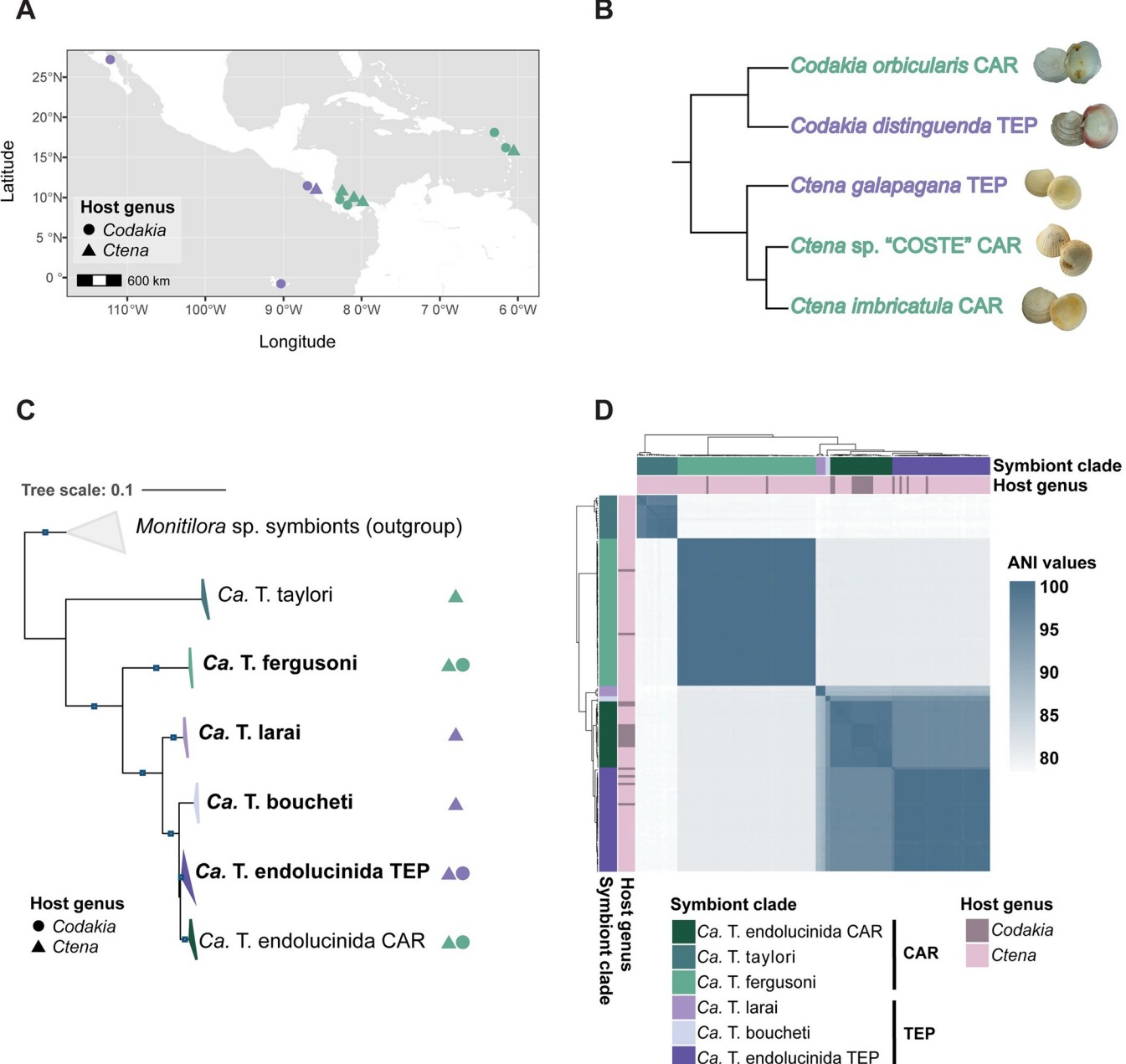

**Fig 1. Phylogenetic relationships of bacterial symbiont clades associated with lucinid sister pairs across the Isthmus of Panama. A** Sister species from the lucinid genera *Codakia* (circles) and *Ctena* (triangles) were sampled on the CAR (Caribbean; turquoise) and TEP (Tropical Eastern Pacific; purple) side of the Isthmus. The map was generated with data from Natural Earth (http://www.naturalearthdata.com/) using the R package "rnaturalearth" (v0.3.2) (https://github.com/ropensci/rnaturalearth). **B** Schematic representation of the phylogenetic relationships of the *Codakia* and *Ctena* species collected, based on the most recent taxonomic study [23]. **C** Six symbiont lineages, four previously undescribed (bold text), were associated with either *Codakia* (circles) or *Ctena* (triangles) hosts across the Isthmus. Maximum likelihood phylogenomic tree of symbiont MAGs recovered from the gills of host sister pairs inferred from GTDB's (Genome Taxonomy Database) multiple sequence alignment using the best fit model Q.plant+F+I+G4. MAGs of *Monitilora ramsayi* symbionts [18] were used as an outgroup. Blue squares indicate ultra fast bootstrap (UFB) values above 95% and SH-aLRT values above 80%. All monophyletic clades were collapsed by location to facilitate interpretation. **D** No symbiont clade was found on both sides of the Isthmus. Heatmap depicting average nucleotide identities (ANI) between the symbiont clades found across the Isthmus. The symbiont clades are colored in tones of green (Caribbean side "CAR") and purple (TEP side) and the genus of the host from which the MAGs were recovered is indicated by the colors violet (*Codakia*) and pink (*Ctena*).

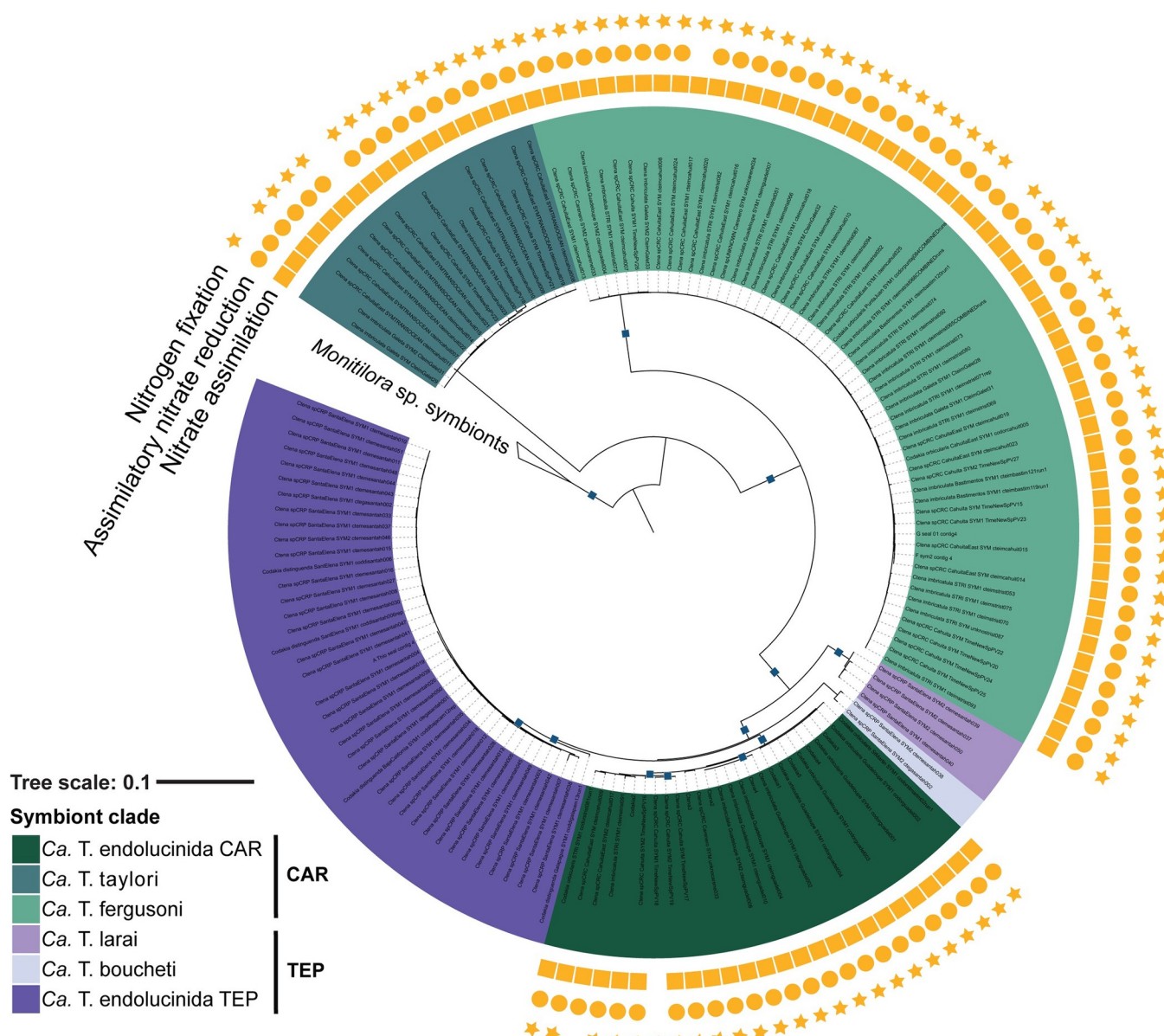

**Fig 2. Metabolic pathways associated with nitrogen acquisition were enriched in the symbiont clades from the Caribbean side of the Isthmus.** Nitrogen fixation (star), assimilatory nitrate reduction (circle) and nitrate assimilation (square) were highly prevalent in the three symbiont clades from the CAR (Caribbean) and absent in all the MAGs extracted from the TEP (Tropical Eastern Pacific). The metabolic pathway enrichment analysis results are superimposed on the phylogenomic tree of the symbionts found across the Isthmus. Blue squares indicate UFB values above 95% and SH-aLRT values above 80%. The symbiont clades are colored in tones of green (Caribbean) and purple (TEP) as in Fig 1C and 1D.

boucheti MAGs encoded both forms I and II, as did the MAGs of closely related *Ca.* T. endolucinida.

The metabolic enrichment analysis across the Isthmus revealed that nitrogen fixation (found in 99% of Caribbean MAGs), nitrate assimilation (which includes nitrate transport), and assimilatory nitrate reduction (both found in 96% of Caribbean MAGs) pathways were enriched in the MAGs of all three symbiont clades found in the Caribbean (Fig 2). All three Caribbean symbiont clades (*Ca.* T. endolucinda CAR, *Ca.* T. taylori, and *Ca.* T. fergusoni) consistently encoded these three nitrogen metabolic pathways that were absent in symbiont

**Table 1. Predicted metabolic functions enriched in the MAGs of symbionts associated with lucinid sister species collected from either side of the Isthmus of Panama.** The annotated functions in this table were present in more than 90% of the MAGs in one group and in less than 10% in the other.

| KOfam | Proportion Caribbean MAGs | Proportion TEP MAGs | Biological function |
|---|---|---|---|
| Nitrite reductase [NAD(P)H] large subunit | 0.9901 | 0 | Assimilatory nitrate reduction |
| Assimilatory nitrate reductase catalytic subunit | 0.9802 | 0 | Assimilatory nitrate reduction |
| Nitrite reductase [NAD(P)H] small subunit | 0.9802 | 0 | Assimilatory nitrate reduction |
| Nitrate/nitrite transport system permease protein | 1 | 0.0213 | Nitrate/nitrite assimilation—transport |
| Nitrate/nitrite transport system substrate-binding protein | 1 | 0.0213 | Nitrate/nitrite assimilation—transport |
| Nitrate/nitrite transport system ATP-binding protein | 0.9802 | 0.0213 | Nitrate/nitrite assimilation—transport |
| Nitrogen fixation protein NifY | 0.9802 | 0 | Nitrogen fixation—biosynthesis and assembly |
| Nitrogen fixation protein NifQ | 0.9802 | 0 | Nitrogen fixation—biosynthesis and assembly |
| Nitrogen fixation protein NifB | 0.9901 | 0 | Nitrogen fixation—biosynthesis and assembly |
| Nitrogenase molybdenum-cofactor synthesis protein NifE | 0.9802 | 0 | Nitrogen fixation—biosynthesis and assembly |
| Nitrogen fixation protein NifZ | 0.9703 | 0 | Nitrogen fixation—biosynthesis and assembly |
| Homocitrate synthase NifV | 0.9703 | 0 | Nitrogen fixation—biosynthesis and assembly |
| Nitrogenase molybdenum-iron protein NifN | 0.9703 | 0 | Nitrogen fixation—biosynthesis and assembly |
| NifU-like protein | 0.9604 | 0 | Nitrogen fixation—biosynthesis and assembly |
| Nitrogenase iron protein NifH | 0.9802 | 0 | Nitrogen fixation—catalytic |
| Nitrogenase molybdenum-iron protein alpha chain NifD | 0.9802 | 0 | Nitrogen fixation—catalytic |
| Nitrogenase molybdenum-iron protein beta chain NifK | 0.9802 | 0 | Nitrogen fixation—catalytic |
| Nif-specific regulatory protein NifL | 0.9901 | 0 | Nitrogen fixation—regulation |
| Nitrogen fixation regulatory protein NifA | 0.9901 | 0 | Nitrogen fixation—regulation |
| ADP-ribosyl-[dinitrogen reductase] hydrolase DraG | 0.9802 | 0.0213 | Nitrogen fixation—regulation |
| NAD+—dinitrogen-reductase ADP-D-ribosyltransferase DraT | 0.9802 | 0 | Nitrogen fixation—regulation |
| Gamma-glutamylaminecyclotransferase | 0.9802 | 0 | Nitrogen fixation—regulation |
| Nitrogen fixation protein NifT | 0.9802 | 0 | Nitrogen fixation—unknown role |
| Nitrogenase-stabilizing/protective protein NifW | 0.9703 | 0 | Nitrogen fixation—structure |
| NADH oxidase ($H_2O$-forming) | 0.9901 | 0 | Response to oxidative stress |
| Electron-transferring-flavoprotein dehydrogenase | 0 | 0.9362 | Electron transfer |
| Electron transfer flavoprotein beta subunit | 0 | 0.9149 | Electron transfer |
| Electron transfer flavoprotein alpha subunit | 0 | 0.9149 | Electron transfer |
| Gamma-polyglutamate synthase CapB | 0.0792 | 0.9574 | Storage compounds |
| Gamma-polyglutamate biosynthesis protein CapC | 0.0792 | 0.9574 | Storage compounds |

MAGs from the TEP, even though *Ca.* T. endolucinida CAR clade was more closely related to clades from the TEP (*Ca.* T. larai, *Ca.* T. boucheti and *Ca.* T. endolucinida TEP) than the two other Caribbean symbiont clades (Fig 2 and S3 Dataset). These enrichment patterns therefore do not correlate with the phylogenetic relationships of the lucinid hosts or their symbionts. The genes enriched in the Caribbean MAGs belonged to the same nitrogen metabolic pathways that were identified through the module enrichment analysis (Table 1 and S3 and S4 Datasets), namely nitrogen fixation and nitrate assimilation. Besides the minimum gene set for nitrogen fixation [28] (*nifHDKENB*), we observed enrichment of a varied repertoire of genes

involved in the process, which included predicted functions in regulation, biosynthesis, assembly and structure. Additionally, a gene annotated as an $H_2O$-forming NADH oxidase was enriched in Caribbean MAGs and was often located in the same genomic region as the nitrogen fixation genes. Both the mapping of the metagenomic reads to the *nif* genes and an HMM search of the nitrogenase against the metagenome assemblies supported the conclusion that the TEP symbionts did not have the potential to fix nitrogen (S5 Dataset). No metabolic modules were found to be enriched in TEP MAGs (Fig 2 and S3 Dataset), but genes encoding an electron-transferring-flavoprotein dehydrogenase and genes involved in gamma-polygluta-mate biosynthesis were enriched in all three symbiont clades from the TEP, a pattern which also did not correlate with symbiont or host phylogeny (Table 1 and S4 Dataset).

## The evolutionary history of nitrogen fixation potential in lucinid symbionts is explained by horizontal gene transfer

**Placing Isthmus symbionts in a global symbiont phylogeny reveals an intermittent distribution of nitrogen fixation genes.** To further investigate the distribution of the nitrogen fixation pathway in lucinid symbionts, we generated a phylogenomic tree combining our newly obtained MAGs with those used in the most recent global tree of lucinid symbionts [18]. In addition, we added 23 new high-quality MAGs from both the *Ca.* Thiodiazotropha and *Sedimenticola* genera (S2 Table). These MAGs originated from specimens of 11 lucinid species collected at various shallow water and low latitude sites to include other oligotrophic and nutrient rich sites beyond the Isthmus (Fig 3A and S1 Table). Seven MAGs clustered within *Ca.* Sedimenticola endophacoides and three MAGs formed a *Sedimenticola* sister clade composed exclusively of symbionts from lucinids of the Pegophyseminae subfamily, which will be provisionally referred to as "PEGO". Others clustered with *Ca.* T. boucheti, *Ca.* T. fergusoni and *Ca.* T. endolucinida. Three MAGs did not form clades with any other MAGs (*Austriella corrugata*, *Ctena bella* Hawaii 21, *Ctena bella* French Polynesia symbionts), and thus represent novel species. The genomic potential for nitrogen fixation was widely, albeit heterogeneously, distributed across the lucinid symbiont tree and found in both *Sedimenticola* and *Ca.* Thiodiazotropha symbionts (Fig 3B). The presence/absence of nitrogen fixation genes varied even within the two species clades *Ca.* S. endophacoides and *Ca.* T. endolucinida. For example, despite their genetic similarity (>95% ANI, S1 Dataset), nitrogen fixation genes were present in the MAGs of the *Ca.* Thiodiazotropha endolucinida lineage associated with *Ctena bella* from Hawaii (*Ca.* T. endolucinida HAW) but not in the *Ca.* Thiodiazotropha endolucinida MAGs retrieved from *Lucina adansonia* from Cape Verde (Fig 3 and S6 Dataset). Similarly, *Ca.* Sedimenticola endophacoides associated with *Phacoides pectinatus* from Florida lacked nitrogen fixation genes, even though this ability was present in closely related lineages of *Ca.* Sedimenticola endophacoides associated with *P. pectinatus* from Guadeloupe (>96% ANI, S1 and S6 Datasets) and Panama (>98% ANI, S1 and S6 Datasets).

**Multiple transfer events account for the sporadic distribution of nitrogen fixation genes amongst the lucinid symbionts.** After dereplication, we analyzed a total of 242 MAGs from the order Chromatiales (including *Ca.* Thiodiazotropha and *Sedimenticola* genera), from which we identified 139 complete *nifHDKT* operons. Trees constructed for each individual *nif* gene had consistent topologies for the strongly supported lucinid symbiont *nif* clades, suggesting that *nifH*, *nifD*, *nifK* and *nifT* are co-inherited (S1 Fig). The resulting *nifHDKT* tree revealed that all the lucinid symbiont *nifHDKT* sequences are closer to each other than to any non-symbiont relative (Figs S2 and 4A). Moreover, our analysis revealed the presence of three distinct major symbiont *nifHDKT* clades, which we have denoted as Clade A, Clade B, and Clade C (Figs S2 and 4A). Clade A comprises genes from *Ca.* Sedimenticola endophacoides.

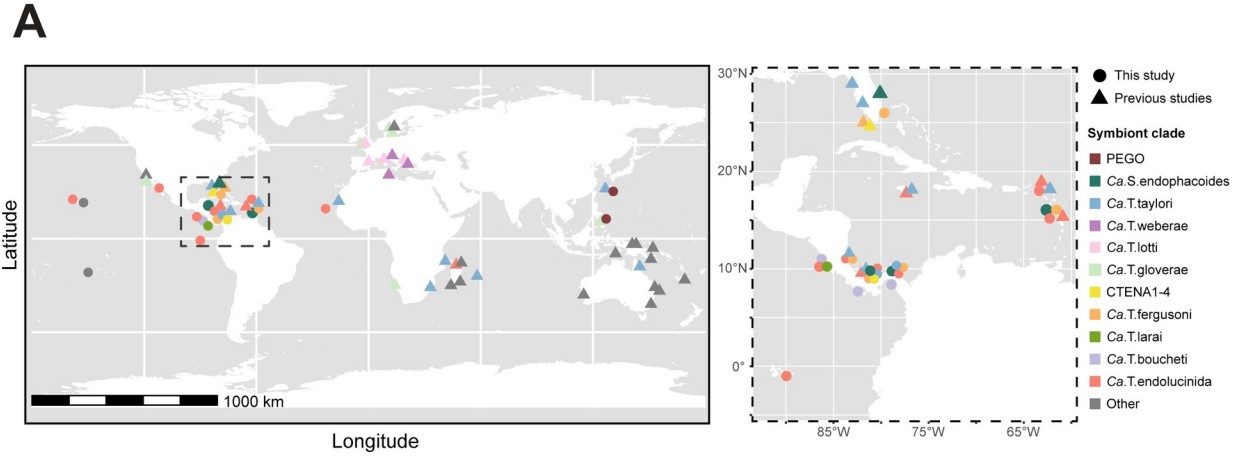

**Fig 3. Placement of new MAGs in the global lucinid symbiont tree reveals discontinuous distribution of nitrogen fixation genes, even within species-level clades. A** Geographic origins of the MAGs included in this analysis. Points were colored based on the clade they belong to. Different shapes indicate whether the samples were obtained in this study (circles) or in previous studies (triangles). The magnified map of the Isthmus—where the density of sampling sites was high—shows these sampling sites in detail. The map was generated with data from Natural Earth (http://www.naturalearthdata.com/) using the R package "rnaturalearth" (v0.3.2) (https://github.com/ropensci/rnaturalearth). **B** Maximum likelihood phylogenomic tree inferred from GTDB's multiple sequence alignment using the best fit model Q.plant+F+I+G4. The names of clades found in the host sister species across the Isthmus of Panama are in turquoise (CAR; Caribbean) or purple (TEP; Tropical Eastern Pacific) font, while globally-distributed symbiont clades are highlighted in gray. Previously described clades were collapsed and annotated in the same way as in the most recent phylogenetic analysis of lucinid symbionts [18] (S2 Table). New clades were collapsed based on ANI (>95%) and/or location. Clades with the potential for nitrogen fixation are indicated with a star and colored symbols match the symbology of the map. Black squares indicate UFB values above 95% and SH-aLRT values above 80%.

Clade B consists of genes from *Ca*. Thiodiazotropha endolucinida, Ctena4, Monit1 (*Monitilora ramsayi* symbionts), and Pegophyseminae symbionts "PEGO". Lastly, Clade C encompasses genes from *Ca*. Thiodiazotropha taylori, lotti, weberae, and fergusoni, as well as Ctena2 and Ctena3 (Fig 4A and 4B). The topology of these clades was inconsistent with the phylogenomic tree (Figs 3B and 4A). Although the *Ca*. T. endolucinida, Ctena4, and Monit1 symbionts belong to the genus *Ca*. Thiodiazotropha, the *nifHDKT* genes from these symbionts formed a clade with the *nifHDKT* genes of the "PEGO", which belongs to the *Sedimenticola* genus (Fig 4A).

We investigated the ancestral states of *nifHDKT* presence or absence to understand the evolutionary processes that could explain the incongruence between the symbiont phylogenomic tree and the *nifHDKT* tree (Fig 4B). According to this analysis, the last common ancestor (LCA) of the Sedimenticolaceae, as well as the LCAs of both *Ca*. Thiodiazotropha and *Sedimenticola*, did not possess the potential to fix nitrogen (Figs S4 and 4B); independent gene gain or loss events explain the sporadic distribution of this metabolic function across the symbiont tree. The last common ancestor (LCA) of *Ca*. Sedimenticola endophacoides was inferred to possess *nifHDKT* genes, indicating a subsequent loss of the nitrogen fixation potential in the Florida symbiont lineage. Conversely, we identified three well-supported instances of *nifHDKT* horizontal gene transfer (Figs 4B and S3) to and from the *Ca*. Thiodiazotropha endolucinida clades. The LCA of the *Ca*. T. endolucinida clade lacked *nifHDKT* genes but the genes were subsequently acquired from an ancestral node of the *Ca*. Sedimenticola "PEGO" lineage before the LCA *Ca*. T. endolucinida HAW and CAR lineages diverged (Figs 4B and S3). Additionally, the LCA of the Ctena4 lineage acquired the ability to fix nitrogen from an ancestral node of the *Ca*. T. endolucinida Hawaii lineage, while a *Monitilora ramsayi* symbiont (Monit1), acquired the *nifHDKT* genes from an ancestral node of *Ca*. T. endolucinida CAR and HAW (Figs S3 and 4B). The ancestral reconstruction and gene reconciliation analyses were, however, unable to resolve the patterns of nitrogenase gene loss and gain across the deep-branching nodes of the other *Ca*. Thiodiazotropha symbiont clades (e.g. the LCA of *Ca*. T. taylori, *Ca*. T. lotti and *Ca*. T. weberae).

## Homologous recombination contributes to cohesion of *Ca*. T. endolucinida populations around the world

To investigate whether homologous recombination might play a role in maintaining the genetic connectivity of the *Ca*. T. endolucinda populations from different geographic locations, we measured the relative rates of recombination to mutation events from core genome alignments of all the *Ca*. T. endolucinda lineages. The R/θ of the *Ca*. T. endolucinda core genome—a 2,710,509 base pairs (bp) alignment—was 0.0999 while the r/m ratio was 0.589 (Table 2), which is significantly higher than the value previously measured for this species (0.082) [19]. This notable difference could be explained by a lack of resolution in the data, as

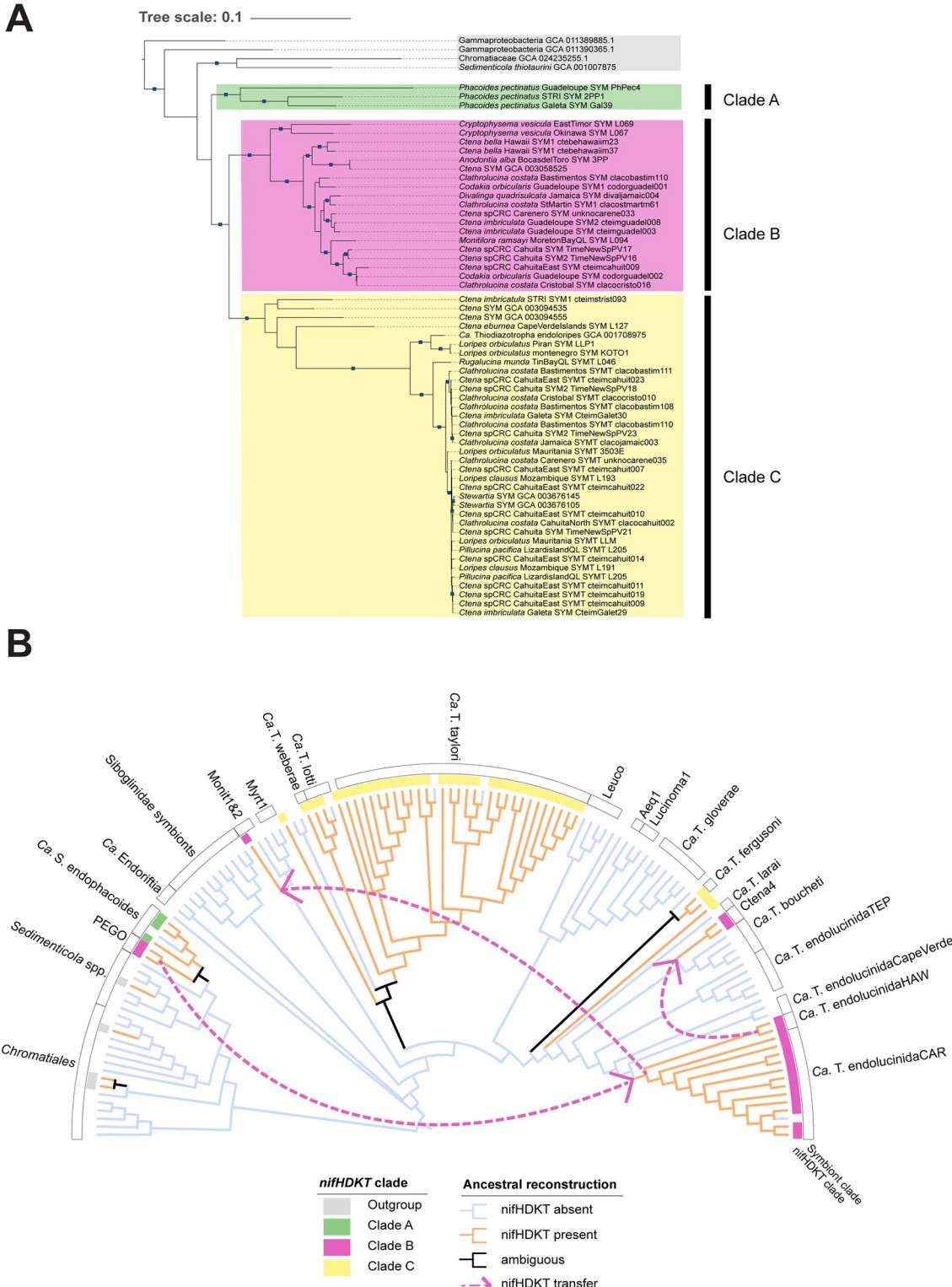

**Fig 4. Horizontal gene transfer events among lucinid symbionts explain incongruence between *nifHDKT* phylogeny and phylogenomic tree. A** The symbiont *nifHDKT* genes formed three major clades that were incongruent with the phylogenetic relationships of the symbionts. Pruned maximum likelihood phylogenetic tree depicting the clades of lucinid symbionts' *nifHDKT genes*. The tree was inferred using the best fit model GTR+F+I+R6 from a concatenated alignment of these genes. **B** The last common ancestor of the *Ca.* Thiodiazotropha genus lacked nitrogen fixation genes, which were subsequently independently acquired through

horizontal gene transfer by different symbiont lineages. Ancestral reconstruction of the presence/absence of *nifHDKT* (orange—*nifHDKT* present, blue—*nifHDKT* absent, black—ambiguous state) mapped onto a pruned cladogram based on a maximum likelihood phylogenomic tree inferred using the best fit model Q.insect+F+R9 from a GTDB's multiple sequence alignment from dereplicated genomes (at 99.5% ANI). Symbiont clades are annotated as in Fig 3B and the clades of their corresponding *nifHDKT* genes are annotated according to A. Robust horizontal gene transfer events inferred from the reconciliation of the gene tree with the phylogenomic tree were superimposed and are depicted as pink arrows.

Osvatic and colleagues analyzed *Ca*. T. endolucinda MAGs from a single site (Bocas del Toro, Panama) [19]. Furthermore, we explored how these rates of homologous recombination compared with those observed in *Ca*. T. gloverae, a globally-distributed symbiont inhabiting deep-water or temperate environments, as well as the rates previously reported for *Ca*. T. taylori, a symbiont found in tropical oligotrophic shallow-water environments around the world [18,19]. The R/θ of the *Ca*. T. gloverae core genome (1,619,656 bp) was 0.121 while the r/m ratio was 1.138 (Table 2). The rates of recombination to mutation events in *Ca*. T. endolucinda was therefore the lowest of the three globally distributed lucinid symbiont species and highest in *Ca*. T. gloverae (Table 2).

## Discussion

### Chemosymbiotic sister species separated by the Isthmus of Panama reveal symbiont adaptation to changing environments

Studies on the adaptations enabling animals to cope with environmental changes resulting from the closure of the Isthmus of Panama (reviewed in [8]) have largely overlooked the role of host-associated microbes [1]. Our investigation focused on understanding how environmental conditions on either side of the Isthmus of Panama have influenced the distribution, diversity and metabolic functions of symbionts associated with lucinid sister species separated by the closure of the Isthmus. *Codakia* and *Ctena* sister species from either side of the Isthmus hosted distinct clades of symbionts from the genus *Ca*. Thiodiazotropha. Despite the absence of symbiont clade overlap across the Isthmus, we observed a lack of specificity between hosts and symbionts, which was evident in the incongruence of their respective phylogenetic relationships. We observed multiple instances where different host species of the *Codakia* and *Ctena* genera on the same side of the Isthmus share the same symbiont groups. This is consistent with horizontal symbiont acquisition from the environment and indicates that the environment is a key factor influencing symbiont selection and distribution [19,29,30].

We compared the metabolic potential of the symbionts from either side of the Isthmus to investigate how the vastly different environmental conditions of the Caribbean Sea (CAR) and TEP have shaped the evolution of lucinid symbionts and their metabolic capabilities. We discovered that nitrogen fixation genes were encoded in the MAGs of all Caribbean symbionts but absent in all the MAGs of symbionts from the TEP. Similarly, we observed that the capacity

**Table 2. Ratios of the rates of homologous recombination to mutation events in globally distributed lucinid symbionts.** Geographic distribution of the clades can be found in S5 Fig.

|  | *Ca*. T. endolucinida Bocas* | *Ca*. T. endolucinida | *Ca*. T. taylori* | *Ca*. T. gloverae |
|---|---|---|---|---|
| **# MAGs** | 18 | 37 | 25 | 12 |
| **Alignment length (bp)** | 3,420,600 | 2,710,509 | 1,751,700 | 1,619,656 |
| **r/m (95% CI)** | 0.082 (0.079–0.084) | 0.587 (0.576–0.598) | 0.814 (0.809–0.819) | 1.138 (1.103–1.184) |

* Values were obtained from Osvatic et al 2021 [19]

for assimilatory nitrate reduction was ubiquitous in symbionts from the Caribbean, but not in those from the TEP (Fig 2). Assimilatory nitrate reductases are similarly absent in many deep-water symbiont lineages, which also lack nitrogen fixation genes [18], suggesting that these two capabilities may be linked. The coastal waters of the TEP are frequently enriched with nutrients due to seasonal upwelling, leading to nitrate levels that are roughly ten times higher than those found in seagrass beds in the Caribbean [31,32]. Hence, the absence of nitrogen fixation genes in the symbiont MAGs from the nitrogen-rich TEP region is consistent with the hypothesis that lucinid symbiont diazotrophy has evolved as an adaptation to life in nitrogen-poor oligotrophic habitats like tropical coral reefs and seagrass beds [33]. Although the TEP symbionts did not possess the capacity to fix nitrogen, the MAGs of all three TEP symbiont lineages encoded unique accessory metabolic capabilities that were lacking in the Caribbean symbionts. Specifically, TEP symbionts had the genetic potential for synthesizing gammapoly-glutamate (Table 1), a storage compound produced by bacteria during nutrient limitation [34]. Electron-transferring-flavoprotein (ETF) dehydrogenase genes were also enriched in the MAGs from the TEP symbionts (Table 1); this enzyme is upregulated in *Pseudomonas aeruginosa* and *Pseudomonas syringae* in response to low temperatures [35,36] and in *Neisseria gonorrhoeae* under anaerobic conditions [37]. The TEP environment is characterized by drastic seasonal changes in physical conditions and nutrient availability. For example, nitrate concentrations in the upper layer of water differ by approximately one order of magnitude between the wet and dry seasons [38–41]. It is intriguing to speculate that the gammapolyglu-tamate synthesis and ETF dehydrogenase genes are beneficial to the TEP symbionts during the seasonal environmental changes typical of upwelling regions, which include changes in nutrient levels, colder temperatures, or reduced oxygen levels [38–41].

## Lucinid symbionts have convergently evolved the ability to fix nitrogen on multiple occasions

We reconstructed the phylogenetic relationships of the *nifHDKT* genes to gain insights into the evolution of nitrogen fixation in the lucinid symbionts and the factors underlying the sporadic distribution of this metabolic capability across the lucinid symbiont tree. While clades A and C of the *nifHDKT* gene tree mirrored the phylogenetic relationships of the symbionts, the incongruence of clade B with symbiont tree topology suggests the *nifHDKT* genes of the *Ca*. T. endolucinida, *Monitilora ramsayi* symbionts (Monit1), and Ctena4 lineages have not co-evolved and/or co-diversified with the single-copy core genes in their respective genomes (Fig 4A). To further investigate this incongruence, we inferred the ancestral states and horizontal gene transfer events of the *nifHDKT* genes throughout the evolution of lucinid symbionts. This analysis indicated that diazotrophy was most likely not an ancestral trait of either the *Sedimenticola* or *Ca*. Thiodiazotropha genera, but was acquired independently by different symbiont lineages that inhabit nutrient-poor environments (Figs S4 and 4B). For example, *nifHDKT* genes were absent in the last common ancestor (LCA) of the *Ca*. T. endolucinida species clade (Fig 4B). Consequently, the TEP, Cape Verde *L. adansoni* and *Cardiolucina* cf. *quadrata* lineages, which either originate from nutrient-rich upwelling regions or deep waters (S2 Table), are incapable of fixing nitrogen, possibly due to the absence of selection pressure that would drive the acquisition and maintenance of this function. Our analysis further predicted that *nifHDKT* genes were later acquired by horizontal gene transfer from a *Sedimenticola* bacterium to the LCA of the *Ca*. T. endolucinida HAW and *Ca*. T. endolucinida CAR lineages. The source of this transfer could have been either an ancestral population of the Pego-physeminae symbionts "PEGO" or an unsampled closely related lineage. Our analyses also indicated a well-supported transfer event from an ancestral population of the *Ca*. T.

endolucinida HAW clade to the Ctena4 clade, which comprises samples from the Florida Keys and the Caribbean (Fig 4B). These findings indicate that both these clades have convergently acquired nitrogen fixation capabilities in an oligotrophic environment, and further suggests the horizontal transfer of nitrogen fixation genes might be a major factor enabling lucinid symbiont adaptation to nitrogen-poor conditions. The third well-supported transfer event was inferred from an ancestral node of the clade consisting of *Ca*. T. endolucinida CAR and HAW to Monit1. Both the Monit1 clade and *nif*-lacking Monit2 clade each consisted of a single *Monitilora ramsayi* symbiont MAG. Given that both MAGs were obtained from samples of the same host species and location (Queensland, Australia) (Fig 3B), and no additional metadata is available, further investigation is required to understand evolution of nitrogen fixation in the *Monitilora ramsayi* symbiosis. The acquisition of *nifHDKT* genes by *Ca*. T. endolucinida from a *Sedimenticola* lineage, rather than one of the *Ca*. Thiodazotropha is unexpected, given that transfers are more likely within the same clade as gene flow tends to occur more frequently among genetically similar bacteria [42]. A possible explanation for this could be the higher abundance of *Sedimenticola* OTUs in sediment bacterial communities [43] compared to *Ca*. Thiodazotropha-like OTUs [44]. This disparity in abundance could reduce the frequency of physical encounters between the different *Ca*. Thiodazotropha lineages—thereby reducing the likelihood of genetic exchange—while increasing the likelihood of encounters with members of the *Sedimenticola* lineage. However, the abundance of free-living "PEGO" symbionts and their close relatives has not been measured, and further investigation of the free-living microbial communities in lucinid habitats is required to gain a more comprehensive understanding of these dynamics. Finally, the ambiguous ancestral states of the deeper branching nodes of the other *Ca*. Thiodazotropha clades hinder our interpretation of gene loss and/or reacquisition events in these symbiont lineages. Filling the gaps in the lucinid symbiont phylogeny by including novel symbiont and/or free-living relative genomes may resolve the uncertainties in the ancestral state reconstruction.

## Nitrogen availability and HGT drive the ecological diversification of lucinid symbionts

The genes for fixing nitrogen were ubiquitous in the genomes of *Ca*. T. endolucinida clades from the oligotrophic waters of the Caribbean and Hawaii but absent in the TEP and Cape Verde clades originating from upwelling regions characterized by sporadically high levels of bioavailable nitrogen [38,45]. Our findings suggest that the ability to fix nitrogen, acquired through HGT of the *nif* operon, likely played a crucial role in altering the ecological niche of the last common ancestor (LCA) of the *Ca*. T. endolucinida CAR and HAW lineages, thus shaping their evolutionary trajectory and divergence. However, all *Ca*. T. endolucinida lineages cross the 95% ANI threshold, widely used to delimit bacterial species, which suggests that these clades, in spite of their essential differences in nitrogen metabolism, represent a single species [24–26]. While ANI by itself does not provide evidence of gene flow between the geographically distinct populations, reconstruction of recombination events within the *Ca*. T. endolucinda species clade further showed that the ratio of the effects of recombination and mutation (0.6) surpassed the theoretical threshold (0.25) necessary to hinder population divergence [46,47]. This suggests there are low barriers to gene flow between geographically separated populations of *Ca*. T. endolucinda and that homologous recombination contributes to the genetic cohesion of the clade. Nevertheless, barriers to genetic exchange or sexual isolation are not prerequisites for ecological and genetic divergence [48]. Indeed, we observed geographic and phylogenetic differentiation of the *Ca*. T. endolucinida populations (Figs 3B and S5) without reproductive isolation (Table 2). A similar phenomenon has been described for

closely-related populations of *Synechococcus* [49] and *Vibrio* [50], for which there is evidence of ecological and genetic divergence, even though these populations also exhibit elevated rates of recombination. Our findings align with these observations, suggesting that during the microbial speciation process, ecological divergence is more likely to precede the emergence of genetic barriers that eventually lead to sexual isolation [51].

The discovery of *Ca*. T. endolucinida populations in the TEP, Hawaii and Cape Verde, substantially increases the distribution range of *Ca*. T. endolucinida, which, with fewer samples, was thought to be restricted to the Caribbean, and also makes this species the third globally-distributed lucinid symbiont from the genus *Ca*. Thiodiazotropha after *Ca*. T. taylori [19], and *Ca*. T. gloverae [18] (S5 Fig). It is also interesting to note that the r/m ratios within *Ca*. T. gloverae nearly double those observed for both *Ca*. T. taylori and *Ca*. T. endolucinida. Furthermore, in contrast to *Ca*. T. endolucinida, there were no major differences in the ability to fix nitrogen across different lineages within *Ca*. T. taylori, and *Ca*. T. gloverae [18,19]. These differences in diversification patterns within the *Ca*. T. taylori, *Ca*. T. endolucinida, and *Ca*. T. gloverae symbiont species, together with their global distribution ranges, present an interesting opportunity to study the origins of genetically cohesive units with species-like properties and characterize the ecological factors underlying bacterial bacterial diversification. For example, further studies on *Ca*. T. endolucinida could provide new insights into how acquiring new genes can drive ecologically important changes in tolerances to physical and chemical conditions (ETF dehydrogenase) or changes in resources consumed (nitrogen metabolism). Conversely, the absence of the ability to fix nitrogen in the *Ca*. Sedimenticola endophacoides clade associated with *P. pectinatus* from a region in Florida that experiences seasonal upwelling events [52–54] presents the opportunity to study how a change in access to abundant nutrient resources could drive gene loss and eventually lead to diversification of this clade.

Our findings highlight the complex interplay between the environment and gene exchange that shapes the evolution and diversification of bacterial symbiont populations. The remarkable flexibility in the partnerships between lucinid clams and diverse symbiont candidates, all sharing the same core metabolic capabilities but different accessory genes, could enhance the resilience of these associations to changing environmental conditions. The capacity to acquire novel metabolic capabilities in response to different nutrient conditions could have important implications for how animal-bacteria symbioses might respond to anthropogenic changes in the environment, such as nutrient loading due to changes in land use. Future investigations should focus on whether location-specific metabolic traits, such as nitrogen fixation, directly influence host fitness or whether symbiont selection is predominantly determined by partner availability.

## Methods

### Sample collection

**Fresh samples.** Live clams were collected with a hand-held trowel in seagrass beds (except for *Phacoides pectinatus* which was collected in mangrove mud) at ~30 cm sediment depth. A colander was used to sieve out the sediment and separate the clams. Gills were dissected directly in the field with a razor blade upon returning to the beach, preserved in DNA/RNA Shield (Cat. No. R1100-250; ZymoBiomics, USA) according to manufacturer's instructions and kept at room temperature during travel and at -20°C for long-term storage. Specimens of *Anodontia alba*, *Codakia orbicularis*, *Codakia distinguenda*, *Clathrolucina costata*, *Radiolucina Jessicae*, *Ctena sp.* COSTE, *P. pectinatus* and *Ctena* cf. *galapagana* from Costa Rica and Panama were collected during the #istmobiome Project sampling campaign (https://istmobiome.net) in 2018 and 2019. Specimens from Guadeloupe were collected by hand in 2019 from seagrass

beds of *Thalassia testudinum* (*Ctena imbricatula*, *C. orbicularis*) and from mangrove mud (*P. pectinatus*) (S1 Table).

**Museum samples.** Specimens of *C. distinguenda*, *Lucinisca fenestrata*, *Lucina adansoni*, *Ctena bella*, *Euanodontia ovum*, *Cryptophysema vesicula* and *Austriella corrugata* were acquired from the collections of the Florida Natural History Museum (FLMNH), Gainesvillae, FL, USA, the California Academy of Sciences in San Francisco, CA, USA and the Natural History Museum (NHM) London, UK (S1 Table). Access was granted and organized by Dr. John Taylor (NHM), Dr. Gustav Paulay and Dr. Amanda Bemis (FLMNH); and Dr. Elizabeth Kools and Dr. Christina Piotrowski (California Academy of Sciences) (S1 Table).

## DNA extraction and sequencing

DNA was extracted from gill tissues using the Qiagen DNeasy Blood and Tissue kit (Cat. No. 69506; Qiagen, USA) following the manufacturer's instructions. Proteinase K digestion of the gills was performed for 48 hours at 56˚ Celsius. Extracted DNA was treated with RNase A (Cat. No. 19101; Qiagen, USA) for 30 minutes at 25˚ Celsius. Paired-end Illumina sequencing produced reads of 150bp or 250 bp in length (S1 Table). Illumina sequencing generated a minimum of 3,000,000 reads per sample. Samples ctemesantah004, cteimcahuit014, and cteimbast119 were also sequenced using PacBio Sequel II long-read technology. One cell was sequenced for each of the three gill samples. For these three samples, only PacBio bins were included for downstream analysis.

## Read quality filtering, assembly, and binning

**Illumina.** Illumina read libraries were trimmed, PhiX contamination filtered, and quality checked using BBMap v37.61's BBDuk feature [55]. Individual read libraries were assembled using SPAdes v3.13.1 [56]. The assembly statistics were assessed with the BBTools (https://sourceforge.net/projects/bbmap/) script "stats.sh". The resulting metagenomic assembly scaffolds were binned with a combination of anvi'o v6.1 [57,58] using CONCOCT v1.1.0 binning [59], and metabat v2.15 [60]. The bins were then compared using dRep v2.4.2's dereplicate workflow [61].

**PacBio.** Samples ctemesantah004, cteimcahuit014, and cteimbast119, for which Illumina MAGs of the symbiont lineages *Ca.* T. endolucinida TEP, *Ca.* T. taylori, and *Ca.* T. fergusoni had been respectively recovered, were selected for long-read sequencing with the goal of obtaining circularized symbiont MAGs of some of the most prevalent clades present across the Isthmus. HiFi reads were produced using circular consensus sequencing (CSS) mode on the PacBio long-read Sequel II system (S1 Table). BBMap v37.61's Seal feature was used to split reads into host and symbiont, based on kmer distributions [55]. A minimum kmer fraction of 0.5 and an input quality offset of 31 eliminating the hamming distance were applied for splitting the reads. MAGs binned out from Illumina paired-end sequencing of 250bp were used as a reference to split host and symbiont reads. Duplicate reads were subsequently removed using BBMap's *reformat.sh* script. Flye v.2.9.2, a *de novo* assembler for single-molecule sequencing reads [62], was used with default parameters and the *-pacbio-hifi* flag to assemble PacBio reads and bin out symbiont genomes. Each Flye assembly resulted in one fully circularized contig that was extracted for downstream analyses (S1 and S2 Tables).

**Bin quality check and taxonomic assignment.** Bins obtained both with Illumina and PacBio sequencing were checked for completion using the CheckM2 v1.0.1 [63] and manually refined using 'anvi-refine'. Bins that were determined to be 90% or more complete and less than 10% contaminated post refinement were considered to be high-quality MAGs [64]. Only high-quality MAGs classified as Gammaproteobacteria by the GTDB-Tk v2.1.1 (Genome

Taxonomy Database Toolkit) classify workflow [65–68] were used for further analyses. MAG depth and breadth of coverage statistics was obtained using CoverM v0.4.0 (https://github.com/wwood/CoverM) by mapping the metagenomes to their corresponding MAGs.

## MAG annotation and metabolic reconstruction

High-quality Gammaproteobacteria MAGs recovered in this study and lucinid symbiont high-quality MAGs published in Osvatic et al. 2023 [18] were functionally annotated through DRAM v1.4.6 [69] using the Kyoto Encyclopedia of Genes and Genomes (KEGG), UniRef90 and PFAM databases. The anvi'o platform (development version) [57] was also used to infer metabolic pathways encoded in the MAGs as an additional strategy for functional annotation. The MAGs were formatted using "anvi-script-reformat-fasta", after which contigs-databases were generated by "anvi-gen-contigs-database". Functional annotations were assigned to the open reading frames of the contigs-databases using the KOfam HMM database of KEGG orthologs [70,71] with "anvi-run-kegg-kofams". The "anvi-estimate-metabolism" (with—module-completion-threshold 0.9) script was used to estimate the presence of complete KEGG modules in each contigs-database.

## Phylogenetics, relatedness and functional potential of symbionts across the Isthmus of Panama

**Symbiont phylogenomics.** A lucinid symbiont tree was inferred from an alignment containing all the high quality Gammaproteobacteria MAGs recovered in this study, and all publicly available lucinid symbionts MAGs included in the phylogenomic tree published in Osvatic et al. 2023 [18], with *Allochromatium vinosum* (GCA_000025485) included as an outgroup. The concatenated alignment of 120 conserved bacterial marker genes from the MAGs was obtained with the GTDB-Tk v2.1.1 classify workflow [65–68]. A maximum likelihood phylogenomic tree was inferred using IQ-Tree v2.2.2.1 [72–74] with auto substitution model detection, 1,000 ultrafast bootstrap (UFB) replicates and 1,000 samples for SH-aLRT branch testing. Nodes with values of UFB greater or equal to 95% and of SH-aLRT greater or equal to 80% were considered to be strongly supported. The tree was visualized, rooted, and annotated with Interactive Tree Of Life (iTOL) [75]. Symbiont clades were collapsed, and lineages were annotated according to the tree published in Osvatic et al. 2023 [18]. Previously undescribed clades were collapsed based on an ANI threshold of 95% and/or sampling location. Additionally, the tree was pruned using Newick Utils v1.6 [76], leaving only leaves which represented symbionts associated with sets of lucinid sister species present across the Isthmus of Panama: *Codakia orbicularis* (Caribbean) and *Codakia distinguenda* (TEP) and *Ctena imbricatula*—*Ctena* sp. (Caribbean) and *Ctena galapagana* (TEP). *Monitilora* sp. symbionts were also left in the pruned tree as an outgroup.

ANI values were calculated with fastANI v1.1 [25]. To visualize the levels of relatedness between the clades present across the Isthmus, the resulting values were used to build a heatmap with the R package pheatmap v1.0.12 [77].

**Functional comparison across the Isthmus.** Anvi'o's pangenomic workflow [57,78] was used for comparing the functional potential of the symbionts across the Isthmus, using anvio's development version. A pangenome database was computed from all the annotated contigs-databases with "anvi-pan-genome" and mcl-inflation parameter set at 6. The functional enrichment analysis [79] was performed on the pangenome to identify genes enriched in the MAGs of symbionts from either side of the Isthmus. For this, MAGs were classified according to their geographic origin (TEP or Caribbean). The enrichment of the modules was computed on the annotated contigs-databases (see section **MAG annotation and metabolic**

**reconstruction**) by using "anvi-compute-metabolic-enrichment". Genes or modules were deemed differentially enriched if they exhibited a presence in over 90% of one group and less than 10% in the other group, while also being present across all clades belonging to their respective groups. The presence and absence of enriched metabolic pathways was plotted on the phylogenomic tree using the Interactive Tree Of Life (iTOL) [75] binary dataset template. To confirm the lack of nitrogen fixation potential in TEP symbionts' genomes and ensure that the genes were not missed in the binning process, the proteins were directly predicted from both Caribbean and TEP metagenome assemblies with Prodigal v2.6.3 [80]. Subsequently, a search was conducted using hmmsearch [81] v3.1b2 with an expectation value (E-value) threshold of 1e-20 against the Pfam model Fer4_NifH (PF00142), which is accessible at https://www.ebi.ac.uk/interpro/entry/pfam/PF00142/curation/. In addition, raw reads from the Caribbean and TEP metagenomes were mapped with bowtie2 v2.5.3 [82] to *nifHDKT* genes extracted from *Ca*. T. fergusoni, taylori, and endolucinida CAR MAGs, and were subsequently quantified.

## Ancestral reconstruction and phylogenetic reconciliation—resolving phylogenetic conflicts

For the reconstruction of ancestral states and the phylogenetic reconciliation, all genomes and MAGs of the order Chromatiales available in GTDB were included in the analysis. Genomes classified as *Desulfuromonas* in GTDB were also included as an outgroup. The GTDB data was accessed on 2024-02-12. The diversity of MAGs was reduced by following the dRep v3.2.2 [61] dereplicate workflow with the S_ANI parameter set to 0.995 and a phylogenomic tree was inferred from the dereplicated MAGs. The methods and parameters used to build the alignment and infer the tree were the same as described in the section **Symbiont phylogenomics** and the resulting tree was rooted with Gotree v0.4.4 [83]. Ancestral states of presence/absence of the nitrogen fixation potential were inferred with PastML [84], applying the marginal posterior probabilities approximation (MPPA). For visualization purposes, the ancestral reconstruction was pruned using Newick Utils v1.6 [76], leaving only the monophyletic clade containing both *Ca*. Thiodiazotropha and *Sedimenticola* symbiont MAGs. The ancestral reconstruction results were visualized with Interactive Tree Of Life (iTOL) [75], where the tree was displayed as a cladogram for clarity.

Sequences of the *nifH*, *nifD*, *nifK* and *nifT* genes were chosen to infer the evolutionary history of nitrogen fixation because they form an operon with the minimum catalytic gene set for the reaction [28]. Sequences were extracted from the anvi'o annotation [57] (described in the section **MAG annotation and metabolic reconstruction**) of the dereplicated MAGs. MAGs that did not contain the full set of genes were excluded from the analysis (S2 Table). Each gene extraction [85] was aligned independently with FSA v1.15.9 [86] using the—fast flag. A tree was inferred from each alignment to ensure that the genes have evolved together. Subsequently, the alignments were concatenated using SeqKit v2.3.0 [87] and a tree inferred from it. The methods and parameters used to infer the trees, define well-supported nodes and visualize the tree were the same as described in the section **Symbiont phylogenomics** and the resulting tree was rooted with Gotree v0.4.4 [83].

The *nifHDKT* tree was reconciled with the dereplicated phylogenomic tree (considered as the species tree) using AleRax v1.0.1 [88] and by employing the UndatedDTL reconciliation model the SPR strategy for gene tree correction and the transfer constrain to parent species. Transfer events observed in the reconciliation, which aligned with a change in the ancestral state from absence to presence of *nifHDKT* (inferred in the ancestral state reconstruction) and

had strong support in the gene tree, were incorporated manually into the ancestral state reconstruction.

## Recombination rates of the bacterial symbionts

We used ClonalFrameML to infer the recombination events across the genomes of globally distributed symbionts based on the workflow described in [19]. To improve computational efficiency, high quality MAGs of *Ca*. T. endolucinda were first de-replicated using dRep v3.2.2 with the S_ANI parameter set to 0.995 [61]. The final sets of de-replicated *Ca*. T. endolucinda MAGs and publicly available *Ca*. T. gloverae MAGs were each separately aligned using the progressiveMauve in Mauve v2.0 [89,90]. Core genomes represented by locally collinear blocks (LCBs) of at least 500 bp were extracted with the Mauve v2.0 command stripSubsetLCBs [90], and then re-aligned using MAFFT v7.304 [91] in auto mode. The LCB alignment was then trimmed with trimAl v1.4 [92] (parameters -resoverlap 0.75 -seqoverlap 80) and a phylogenetic tree was generated using FastTree v2.1.11 [93,94] using the parameters -gtr -nt. The alignment and tree were used as input for ClonalFrameML [95], which was run with 100 (-emsim) pseudo-bootstrap replicates. Recombination events were visualized in R v4.2.3 using the 'cfml_results.R' script available at https://github.com/xavierdidelot/ClonalFrameML; date of accession: 1 September 2023).

## Maps and visualization

Maps were plotted in R using the following packages: rnaturalearth v0.3.2 [96], ggspatial v1.1.7 [97] and ggplot2 v3.4.1 [98]. Final modifications for publication of all figures were done in Adobe Illustrator (https://adobe.com/products/illustrator) to improve readability while preserving the original information from the different methods.

## Supporting information

**S1 Table. Metadata of metagenomes obtained in this study.**
(XLSX)

**S2 Table. Metadata of MAGs used in this study.**
(XLSX)

**S1 Dataset. ANI values of all MAGs versus all MAGs.**
(XLSX)

**S2 Dataset. Metabolic reconstruction of new symbiont clades.**
(XLSX)

**S3 Dataset. Symbiont's functional potential across the Isthmus: Anvi'o's module enrichment analysis.**
(XLSX)

**S4 Dataset. Symbiont's functional potential across the Isthmus: Anvi'o's gene enrichment analysis.**
(XLSX)

**S5 Dataset. Read mapping to *nifHDKT* and nitrogenase HMM search result.**
(XLSX)

**S6 Dataset. Presence and absence of nitrogen fixation potential in symbiont MAGs.**
(XLSX)

**S1 Fig. Maximum likelihood phylogenetic trees of different nitrogen fixation genes of lucinid symbionts. A** *nifH* gene tree **B** *nifD* gene tree **C** *nifK* gene tree **D** *nifT* gene tree. (TIF)

**S2 Fig. Complete maximum likelihood *nifHDKT* phylogenetic tree of the order Chromatiales.** (TIF)

**S3 Fig. Gene-species tree reconciliation of *nifHDKT* operon of lucinid symbionts obtained with AleRax.** (TIF)

**S4 Fig. Complete ancestral reconstruction of the order Chromatiales.** (TIF)

**S5 Fig. Geographic distribution of global symbionts.** The map was generated with data from Natural Earth (http://www.naturalearthdata.com/) using the R package "rnaturalearth" (v0.3.2) (https://github.com/ropensci/rnaturalearth) (TIF)

## Acknowledgments

*Codakia orbicularis* individuals from Florida were collected and sent to us by Dr. Diana Chin (codorflorid129run1). *Divalinga* sp. Individuals from Jamaica were collected and sent to us by Dr. Amber Stubler (divaljamaic004). Part of the sequencing was carried out by the DNA Technologies and Expression Analysis Core at the UC Davis Genome Center and at the Joint Microbiome Facility (JMF) of the Medical University of Vienna and the University of Vienna (project IDs JMF-1911-9, JMF-2104-13, and JMF-2002-8). We thank Petra Pjevac and Gudrun Kohl of the JMF for processing the samples. We are grateful to the Life Science Computer Cluster at the University of Vienna for the computational resources used for parts of the analyses. We also want to specifically thank Minor Lara's sons Minor and Steven for their participation during fieldwork in Cuajiniquil. We would like to acknowledge the use of ChatGPT, an AI language model, for its assistance in correcting grammar and enhancing the clarity of the writing in this manuscript.

**Collection permits:** Sampling in Panama was performed complying with the Panama-Nagoya regulations under the identifier ABSCH-IRCC-PA-254919-1; collection and export were performed under the collection permit SE/AO-4-19 and export permit SEX/A-22-2020. Collection in Costa Rica, and following export and sequencing, were performed under permits R-004-2019-OT-CONAGEBIO and R-017-2022-OT-CONAGEBIO from the Comisión Nacional para la Gestión de la Biodiversidad (CONAGEBIO). Collection in Guadeloupe was performed under permit number TREL2302365S/676.

## Author Contributions

**Conceptualization:** Isidora Morel-Letelier, Benedict Yuen, A. Carlotta Kück, Yolanda E. Camacho-García, Jillian M. Petersen, Minor Lara, Matthieu Leray, Jonathan A. Eisen, Jay T. Osvatic, Olivier Gros, Laetitia G. E. Wilkins.

**Data curation:** Isidora Morel-Letelier, Benedict Yuen, A. Carlotta Kück, Yolanda E. Camacho-García, Minor Lara, Matthieu Leray, Jay T. Osvatic, Olivier Gros, Laetitia G. E. Wilkins.

**Formal analysis:** Isidora Morel-Letelier, Benedict Yuen, A. Carlotta Kück, Jay T. Osvatic, Laetitia G. E. Wilkins.

**Funding acquisition:** Jillian M. Petersen, Matthieu Leray, Jonathan A. Eisen, Laetitia G. E. Wilkins.

**Investigation:** Isidora Morel-Letelier, Benedict Yuen, Yolanda E. Camacho-García, Jillian M. Petersen, Matthieu Leray, Jonathan A. Eisen, Olivier Gros, Laetitia G. E. Wilkins.

**Methodology:** Isidora Morel-Letelier, Benedict Yuen, A. Carlotta Kück, Minor Lara, Jay T. Osvatic, Laetitia G. E. Wilkins.

**Project administration:** Isidora Morel-Letelier, Benedict Yuen, Matthieu Leray, Laetitia G. E. Wilkins.

**Resources:** Isidora Morel-Letelier, Yolanda E. Camacho-García, Minor Lara.

**Supervision:** Benedict Yuen, Yolanda E. Camacho-García, Olivier Gros, Laetitia G. E. Wilkins.

**Validation:** Isidora Morel-Letelier, Benedict Yuen, Jillian M. Petersen, Laetitia G. E. Wilkins.

**Visualization:** Isidora Morel-Letelier.

**Writing – original draft:** Isidora Morel-Letelier, Benedict Yuen, A. Carlotta Kück, Yolanda E. Camacho-García, Jillian M. Petersen, Minor Lara, Matthieu Leray, Jonathan A. Eisen, Jay T. Osvatic, Olivier Gros, Laetitia G. E. Wilkins.

**Writing – review & editing:** Isidora Morel-Letelier, Benedict Yuen, A. Carlotta Kück, Yolanda E. Camacho-García, Jillian M. Petersen, Minor Lara, Matthieu Leray, Jonathan A. Eisen, Jay T. Osvatic, Olivier Gros, Laetitia G. E. Wilkins.

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
