## [Decision Letter · Decision Letter 0]

3 Feb 2024

Dear Dr Wilkins,

Thank you very much for submitting your Research Article entitled 'Adaptations to nitrogen availability drive ecological divergence of chemosynthetic symbionts' to PLOS Genetics.

The manuscript was fully evaluated at the editorial level and by independent peer reviewers. The reviewers appreciated the attention to an important problem, but raised some substantial concerns about the current manuscript. Based on the reviews, we will not be able to accept this version of the manuscript, but we would be willing to review a much-revised version. We cannot, of course, promise publication at that time.

If you decide to revise the manuscript for further consideration at PLOS Genetics, please aim to resubmit within the next 60 days, unless it will take extra time to address the concerns of the reviewers, in which case we would appreciate an expected resubmission date by email to plosgenetics@plos.org.

We are sorry that we cannot be more positive about your manuscript at this stage. Please do not hesitate to contact us if you have any concerns or questions.

Yours sincerely,

Jin Sun

Guest Editor

PLOS Genetics

Kelly Dyer

Section Editor

PLOS Genetics

Dear authors,

Both reviewers have gone through your manuscript. Overall, they find your manuscript interesting, but have also raised some technical issues. I agree with them, espeically on the point of whether the LCA of of Ca. Thiodiazotropha lacked nif gene. Please revise the manuscript according to the suggestions, and i look forward to the revision.

Reviewer's Responses to Questions

**Comments to the Authors:**

Reviewer #1: See attachment

Reviewer #2: The environment is an important driving force for the evolution of bacterial symbionts, especially for the horizontally transferred endosymbionts. In the current study, the authors forced on the sister species of two lucinid genera, Codakia and Ctena, distributed in distinct environmental conditions in the Tropical Eastern Pacific and Caribbean to reveal the influence of the environment in the genomic and metabolic divergence of endosymbionts. Besides, the authors also revealed convergent evolution of nitrogen fixation genes in different evolutionary lineages under nutrient-poor conditions by comparing different lucinid symbionts across the globe. Overall, the manuscript is well-organized, and the results are thoroughly discussed. The authors provided valuable resources on the symbionts of Lucinidae and contributed to the understanding of the evolution and adaption of marine symbiosis. Therefore, I would like to suggest that the manuscript be published after minor revision.

Detailed comments to the authors:

Lines 174-178, different numbers of MAGs were assembled from the individuals of the two lucinid genera, Codakia and Ctena, and significantly more MAGs were assembled from Ctena individuals. Are the differences in the number of assembled genomes related to the sequencing methods or to host species?

In Fig. 1B-C and the relative text in the part of results, I did not find the descriptions of the host species and their corresponding symbiont clades, although they can be found in the S1_table. This information will help evaluate the specificity between the host species and symbiont clades. Therefore, I suggested making it clear about the result in the main text.

In Fig. 1D, the host genus and the symbiont clade were incorrectly annotated at the top of the heatmap.

Line 222, only three symbiont clades were found in the S2 Dataset, the data of other clades were missed in the table.

Line 394, I do not fully agree with the conclusion that “symbiont choice was driven by geographic location rather than host species”. The two host species in genera Codakia hold two clades of Ca.T.endolucinida which is phylogenetically close and has higher ANI values. Besides, the number of symbiotic bacteria obtained from the two lucinid genera was different. In particular, in Ctena individuals, three different clades of symbiotic bacteria were obtained. Are these bacteria coexisted in the gill cells of the host? Are there any previous studies? The environment is responsible for the differentiation of endosymbionts, but the host species is also relevant because the establishment of endosymbiosis requires complex molecular mechanisms in the host cells.

Lines 522-541, the sample collection part is unclear to me. The authors listed many samples, but it is unclear which was used for sequencing in the manuscript. I think it is better to clearly describe which individuals were newly acquired or sequenced in the current study and which were previously obtained by the authors or other studies, and references should be listed if they have been published.

Additionally, S1_table has a similar problem. All the data used in the manuscript were put together in the table. The distinction between newly generated data and data from the previous studies is not clear. In the column of the Paper, the references to the previous studies should be correctly cited.

**Have all data underlying the figures and results presented in the manuscript been provided?**

Reviewer #1: Yes

Reviewer #2: None

PLOS authors have the option to publish the peer review history of their article (what does this mean?). If published, this will include your full peer review and any attached files.

Reviewer #1: No

Reviewer #2: No

---

## [Decision Letter · Decision Letter 1]

8 May 2024

Dear Dr Wilkins,

We are pleased to inform you that your manuscript entitled "Adaptations to nitrogen availability drive ecological divergence of chemosynthetic symbionts" has been editorially accepted for publication in PLOS Genetics. Congratulations!

Yours sincerely,

Jin Sun

Guest Editor

PLOS Genetics

Kelly Dyer

Section Editor

PLOS Genetics

Comments from the reviewers (if applicable):

Reviewer's Responses to Questions

**Comments to the Authors:**

Reviewer #1: Summary: The authors did a wonderful job of addressing my concerns in round 1 through performing some extra analyses, providing more data, and clarifying the text. My concerns about gene content are assuaged by the completed circular assemblies and extra read-based fishing for the incomplete assemblies. I also greatly appreciate the extended ancestral state reconstruction analysis the authors performed with extra taxa to resolve some of the deeper nodes.

Major Comments: None

Minor Comments:

Please include a bit of the explanations you provided in your response to reviewers in the text about: 1) How the ANI approach you used works (so the reader doesn’t have to go look up fastANI’s protocol for bi-directional searches that requires all samples to contain the sequence (i.e., no missing data)). 2) Why the Q.plant model in IQ-tree is an acceptable model for a bacterial genome (so the reader doesn’t have to peruse IQ-Tree forums to find the answer). Brief, one-sentences summaries that cover the main concerns would be sufficient.

Reviewer #2: The authors have made intensive revisions to their manuscript in line with all the comments, providing additional information and analysis. I appreciate their efforts and recommend publishing in PLOS Genetics.

**Have all data underlying the figures and results presented in the manuscript been provided?**

Reviewer #1: Yes

Reviewer #2: Yes

PLOS authors have the option to publish the peer review history of their article (what does this mean?). If published, this will include your full peer review and any attached files.

Reviewer #1: No

Reviewer #2: No

**Data Deposition**

http://datadryad.org/submit?journalID=pgenetics&manu=PGENETICS-D-23-01271R1

**Press Queries**

---

## [Editor Report · Acceptance letter]

27 May 2024

PGENETICS-D-23-01271R1 

Adaptations to nitrogen availability drive ecological divergence of chemosynthetic symbionts 

Dear Dr Wilkins, 

We are pleased to inform you that your manuscript entitled "Adaptations to nitrogen availability drive ecological divergence of chemosynthetic symbionts" has been formally accepted for publication in PLOS Genetics! Your manuscript is now with our production department and you will be notified of the publication date in due course.

With kind regards,

Anita Estes

PLOS Genetics

On behalf of:
